# Integration of mRNA and miRNA Profiling Reveals Heterosis in *Oreochromis niloticus* × *O. aureus* Hybrid Tilapia

**DOI:** 10.3390/ani12050640

**Published:** 2022-03-03

**Authors:** Wei Xiao, Binglin Chen, Jun Wang, Zhiying Zou, Chenghui Wang, Dayu Li, Jinglin Zhu, Jie Yu, Hong Yang

**Affiliations:** 1Key Laboratory of Freshwater Aquatic Genetic Resources, Ministry of Agriculture, National Demonstration Center for Experimental Fisheries Science Education, Shanghai Engineering Research Center of Aquaculture, Shanghai Ocean University, Shanghai 201306, China; xiaowei@ffrc.cn (W.X.); wangjun@shou.edu.cn (J.W.); 2Freshwater Fisheries Research Center, Chinese Academy of Fishery Sciences, Wuxi 214081, China; chenbinglin@ffrc.cn (B.C.); zouzy@ffrc.cn (Z.Z.); lidy@ffrc.cn (D.L.); zhujl@ffrc.cn (J.Z.); yujie@ffrc.cn (J.Y.)

**Keywords:** dominance, heterosis, growth, hybrid tilapia, survival rate, overdominance

## Abstract

**Simple Summary:**

Hybrid tilapia (*Oreochromis niloticus* ♀ × *O. aureus* ♂) are commercially important in China. Because cultured hybrids have superior growth and survival rate relative to the parental stocks, parent species potentially represent model taxa to analyze the molecular mechanisms of hybrid vigor. Comparisons of the growth and hematologic biochemical characters and transcriptional analyses of mRNA and miRNA, approximately 21-nt-long noncoding RNAs that negatively regulate gene expression at the post transcriptional level, were performed in hybrid and parental tilapia stocks to investigate the underlying molecular basis for heterosis. The present study indicated that dominance and overdominance models are widespread in transcriptional and post-transcriptional regulation of genes involved in growth, metabolism, immunity, and antioxidant capacity in hybrid tilapia, thus providing new insights into molecular heterosis in hybrid tilapia and advancing our understanding of the complex mechanisms involved in this phenomenon in aquatic animals.

**Abstract:**

Heterosis is a widespread biological phenomenon in fishes, in which hybrids have superior traits to parents. However, the underlying molecular basis for heterosis remains uncertain. Heterosis in growth and survival rates is apparent in hybrid tilapia (*Oreochromis niloticus* ♀ × *O.* *aureus* ♂). Comparisons of growth and hematological biochemical characteristics and mRNA and miRNA transcriptional analyses were performed in hybrid and parents tilapia stocks to investigate the underlying molecular basis for heterosis. Growth characteristics and hematological glucose and cholesterol parameters were significantly improved in hybrids. Of 3097 differentially expressed genes (DEGs) and 120 differentially expressed miRNAs (DEMs) identified among three stocks (*O. niloticus*, *O. aureus*, and hybrids), 1598 DEGs and 62 DEMs were non-additively expressed in hybrids. Both expression level dominance and overdominance patterns occurred for DEGs and DEMs, indicating that dominance and overdominance models are widespread in the transcriptional and post-transcriptional regulation of genes involved in growth, metabolism, immunity, and antioxidant capacity in hybrid tilapia. Moreover, potential negative regulation networks between DEMs and predicted target DEGs revealed that most DEGs from miRNA-mRNA pairs are up-regulated. Dominance and overdominance models in levels of transcriptome and miRNAome facilitate the integration of advantageous parental alleles into hybrids, contributing to heterosis of growth and improved survival. The present study provides new insights into molecular heterosis in hybrid tilapia, advancing our understanding of the complex mechanisms involved in this phenomenon in aquatic animals.

## 1. Introduction

Heterosis is a phenomenon in which hybrids exhibit phenotypic characteristics superior to those of their parents, such as in growth, adaptability, stress tolerance, and yield and quality [1]. Because of its economic and applied value, it has been widely used to improve phenotypic traits in crops and husbandry [2,3]. Similarly to maize and rice, interspecific reproductive isolation in fishes is not strict, for which reason hybrid vigor has been exploited for decades [4,5]. Through crosses between species and stocks of fishes, complementary advantageous traits can be developed, improving the prospect of hybrid survival and production in aquaculture [6,7].

At present, dominance, overdominance and epistasis are three common hypotheses advanced to explain heterosis [8,9,10], and each has evidential support. However, these hypotheses are also largely conceptual, and they cannot explain the molecular principles of trait expressions. Because hybrids produce no new genes, hybrid vigor is likely a result of quantitative changes in gene expression. Recent developments in genomics have revealed that alleles from parents can interact with each other and alter the expression of genes associated with phenotypes in hybrids, which may produce a super-parental phenotypic advantage [11]. Complemented by developments in molecular biology, transcriptional profiling tools such as cDNA microarrays and RNA-sequencing technologies have enabled the mechanisms of hybrid vigor to be investigated at a genome-wide expression level in fishes. Regulation of cis- and trans-acting elements and the non-additive expression of genes (including non-coding sRNAs) in hybrids may produce global transcriptional differences between fish hybrids and their parents [12,13,14,15,16], resulting in heterosis in hybrid phenotypes.

Hybrid tilapia (*Oreochromis niloticus* ♀ × *O. aureus* ♂) are commercially important in China [17]. Because cultured hybrids have superior growth and survival rates compared to parental stock and are approximately 100% male under careful broodstock management (preferred in culture because males grow faster and larger than females), parental species potentially represent model taxa to analyze the molecular mechanisms of hybrid vigor [18]. Previous studies provided new insights into the overall expression mechanisms of genes such as *gh* and *igf1* in hybrid tilapia [19,20], although functional gene regulation mechanisms associated with hybrid vigor remain unclear. Further study is required to understand the regulation of gene networks that produce superior traits in hybrid tilapia. Several integrated mRNA and miRNA transcription analyses have disclosed complex gene regulatory networks in hybrid plants [11,21], vital for heterosis formation. Few studies, however, have reported these in fishes because of a lack of corresponding fish genome sequence and annotation data. Because the entire tilapia genome has been sequenced [22] and is available, we performed comparative mRNA and miRNAomics analyses on three tilapia stocks, *O. niloticus*, *O. aureus*, and hybrids, to investigate the molecular regulatory mechanisms that determine phenotypic heterosis in hybrids.

## 2. Materials and Methods

### 2.1. Ethics Statement

All experimental protocols were approved by the Animal Care and Use Committee of the Shanghai Ocean University (Shanghai, China). Fish were maintained in well-aerated water, were anesthetized with MS-222 before sampling, and their viscera were extracted in accordance with Guidelines for the Care and Use of Laboratory Animals in China.

### 2.2. Experimental Fish Artificial Breeding and Culture

Adult *O. niloticus* (NL) and *O. aureus* (AR) were obtained from the Wuxi breeding base, Freshwater Fisheries Research Centre, Chinese Academy of Fishery Sciences. Males and females with full body shape and bright body color were selected from the NL and AR tilapia stocks and acclimated in a 500 L glass tank for 2 weeks, with water temperature maintained at 27–28 °C, dissolved oxygen > 6 mg/L, and ammonia nitrogen < 0.3 mg/L. All fish were fed a commercial diet (32% crude protein, Nanjing ADM Animal Nutrition Co., Ltd., Nanjing, China) rich in vitamin E (0.03%), before being intraperitoneally injected with mixed hormones (female: 2500 IU/kg human gonadotropin + 5 mg/kg domperidone + 10 ug/kg luteinizing hormone-releasing hormone A_2_; male: half the dose of the female) (Ningbo Animal Hormone Factory, Ningbo, China). Eight hours after injection, eggs and sperm were collected from females and males by gently squeezing the abdomen. Stocks AR ♀ × AR ♂ (hereinafter referred to as AR), NL ♀ × NL ♂ (hereinafter referred to as NL), and NL ♀ × AR ♂ (hereinafter referred to as HY) were artificially fertilized. Eggs were added into separate Petri dishes and fertilized with sperm, after which water was added and the eggs were stirred for 2 min. Fertilized eggs (AR, NL, HY) were poured directly into an incubator and maintained for 7 days at 28 °C to obtain fry that swam typically.

From each stock 500 healthy fry were randomly selected, placed into 1200 L glass tanks for temporary rearing, and fed homemade powdered feed (43% crude protein, 20 μm diameter) three times daily. After 30 days, 150 males (identified by examining their external reproductive pores) from each stock were selected, weighed using an electric balance, and had passive integrated transponder (PIT) electronic markers injected into their abdomens. From each stock, 50 tilapia were then assigned into each of three 120 m^3^ ponds, with water temperature 28–32 °C, about 13 h of natural daylight per day, an ammonia nitrogen concentration of 0–0.5 mg/L, nitrite concentration of 0–0.3 mg/L, pH of 7.0–8.0, and dissolved oxygen > 5.0 mg/L. During culture, artificial floating feed (30% crude protein, Nanjing ADM Animal Nutrition Co., Ltd., Nanjing, China) of 2 mm diameter was fed daily; after fry were not observed to eat, in ≈ 30min, the excess feed was removed.

### 2.3. Statistical Analyses and Sample Collection

On day 0, 45 and 90 of culture, all fish in the ponds were caught, PIT markers were read, and fish body weight, length and depth were measured. Growth characteristics were calculated and evaluated using the following formulae:Weight Gain Rate (WGR, %) = (*W_t_* − *W_0_*)/*W_0_* × 100
Body Length Gain Rate (BLGR, %) = (*L_t_* − *L_0_*)/*L_0_* × 100
Body Depth Gain Rate (BDGR, %) = (*D_t_* − *D_0_*)/*D_0_* × 100
Specific Growth Rate (SGR, %/day) = (Ln*W_t_* − Ln*W_0_*)/*t* × 100
where *W_t_*, final body weight; *W_0_*, initial body weight; *L_t_*, final body length; *L_0_*, initial body length; *D_t_*, final body depth; *D_0_*, initial body depth; and *t*, experimental days.

After 45 days, 6 individuals were randomly captured from each stock in each pond (18 individuals in total per stock) to investigate differences in hematological parameters and gene transcription levels. After mild anesthesia with 50 mg/L MS-222, 500 μL blood was collected from the tail vein. Serum was separated by centrifuging blood samples (5000× *g*) for 10 min, and analyzed using a BS-600 Automatic Biochemical Analyzer (Mindray, Shenzhen, China). Measured parameters included triglycerides, cholesterol, glucose, alanine aminotransferase (ALT), aspartate aminotransferase (AST), and total proteins (TP). All 18 individuals in each stock were dissected, and 200 mg of liver was removed and frozen in liquid nitrogen for mRNA and miRNAomics sequencing. These analyses were repeated on fish at day 90. The experimental design is depicted in Figure 1.

### 2.4. Total RNA Extraction and Quantitation

Total RNA in liver tissue of each tilapia was isolated using TRIZOL reagent (Invitrogen, Waltham, MA, USA) following manufacturer’s instructions. Genomic DNA was removed by DNase-I (TaKaRa, Shiga, Japan). Nanodrop-2000 (Thermo Fisher, Nockville, MD, USA) was used to measure RNA concentration. RNA from the 6 fish livers from each stock and pond was pooled in equal proportions as a replicate, so for each stock (AR, NL, HY) samples included three replicates, defined as AR1-3, NL1-3, and HY1-3, respectively. RNA degradation and contamination were monitored on 1% agarose gels, and integrity was assessed using a 2100 Bioanalyzer (Agilent, Santa Cruz, CA, USA). Only high-quality RNA samples (OD 260/280 = 1.8–2.2, OD 260/230 > 2.0, RIN > 7) were used to construct a sequence library; 28S/18S > 1.0 cDNA libraries and sRNA libraries were constructed.

### 2.5. cDNA Library Construction and mRNA Sequencing

An RNA-seq transcriptome library was prepared using a TruSeq^TM^ RNA sample preparation Kit from Illumina (San Diego, CA, USA) and 10 μg of total RNA. Messenger RNA was enriched and isolated with oligo (dT) magnetic beads and then randomly cut into small fragments using a fragmentation buffer. Double-stranded cDNA libraries were constructed using a SuperScript cDNA synthesis kit (Invitrogen, Waltham, MA, USA). For mRNA sequencing, nine cDNA libraries were constructed. Libraries were size-selected for cDNA target fragments of 200–300 bp on 2% agarose gels, followed by PCR amplification for 15 cycles, with paired-end RNA sequencing libraries then sequenced on the NovaSeq 6000 platform (2 × 150 bp read length, Illumina, San Diego, CA, USA) at Majorbio (Shanghai, China).

### 2.6. Transcriptome Sequence Alignment, Assembly, and Functional Annotation

Generated raw reads were first filtered by removing adaptors, ambiguous reads (with >5% ambiguous nucleotides), and low-quality reads (containing >20% bases whose quality scores < 10). Clean reads were separately aligned to the reference genome (https://www.ncbi.nlm.nih.gov/assembly/GCA_001858045.3, 21 April 2021, GCF_001858045.2) with orientation mode using HISAT2 (https://ccb.jhu.edu/software/hisat2/index.shtml, 21 April 2021, version 2.1.0) software [23]. Mapped reads of each sample were assembled using StringTie (https://ccb.jhu.edu/software/stringtie/index.shtml?t=example, 21 April 2021, version 2.1.2) [24]. Obtained transcriptomic contigs and unigenes were then aligned to the NCBI non-redundant protein database (NR) and Kyoto Encyclopaedia of Genes and Genomes (KEGG) database.

### 2.7. Small RNA Library Construction and miRNA Sequencing

The nine replicate RNA samples (three from each stock) were used to construct nine small RNA libraries for small RNA sequencing, following the Next^®^ Multiplex Small RNA Library Prep Kit (NEB, Ipswich, MA, USA) manufacturer’s instructions. The nine sRNA libraries were sequenced on the Hiseq 2500 platform (2 × 75 bp read length, Illumina, San Diego, CA, USA) at Majorbio (Shanghai, China).

### 2.8. Identification of miRNAs

Generated raw reads were first trimmed of low-quality bases (quality scores < 20) at the 3′ end using in-house Perl scripts, then sequencing adapters were removed using FASTTX-Toolkit software (http://hannonlab.cshl.edu/fastx_toolkit/, 20 April 2021, version 0.0.14). All identical sequences of 18–32 nt were counted and eliminated from the initial data set. Clean reads were aligned to the RNA family database (http://rfam.sanger.ac.uk/, 20 April 2021, version 12.3) using Bowtie to remove non-miRNA sequences (rRNA, tRNA, snoRNA, etc.) [25]. Known miRNA sequences were identified by aligning clean reads to the miRbase database (http://www.mirbase.org/, 21 April 2021, version 22.1) using BLAST (https://blast.ncbi.nlm.nih.gov/blast.cgi, 21 April 2021, version 2.9.0). Novel miRNA sequences were predicted using miRDeep2 [26].

### 2.9. Differential Expression Analysis and mRNAs and miRNAs

To identify differentially expressed genes (DEGs) and miRNAs (DEMs) among stocks, the expression level of each transcript was calculated according to the number of transcripts per million reads (TPM) method [27]. Essentially, differential expression analysis was performed on the TPM of genes and miRNAs in parents and hybrids using DESeq2 [28]. DEGs between any two stocks were determined to significantly differ (*p* < 0.01) in expression level if there was a fold change > 2 or <0.5. DEMs between two stocks differed significantly in expression level at *p* < 0.05. All DEGs and DEMs between parental stocks were examined to assess their mode of transcriptional regulation in hybrids according to Bougas et al. [29]. Each DEG/DEM was classified as a non-additive gene/miRNA (NEG/NEM) if there was a significant difference in expression levels between hybrids and the average value of parents, or as an additive gene if there was not [30]. KEGG pathway analysis was performed on NEGs using the hypergeometric test by KOBAS [31]. All DEGs and DEMs in the HY stock (compared to NL and AR parents) were divided into 12 possible classes of differential expression [32]: additivity (I, II), expression level dominance by NL (ELD-NL: III, IV), expression level dominance by AR (ELD-AR: V, VI), and expression level overdominance (ELOD-Up: VII–IX; and ELOD-Down: X–XII).

### 2.10. MiRNA Target Gene Prediction and miRNA–mRNA Interaction

Target genes of the DEMs were identified by miRanda5 (http://www.miranda.org/, 21 April 2021, version 3.3a) and RNAhybrid (http://bibiserv.techfak.uni-bielefeld.de/rnahybrid/, 21 April 2021, version 2.1.2) following methodologies in Rehmsmeier [33] and Yang et al. [34]. To investigate possible miRNA-mRNA interaction pairs in HY individuals, the target genes of DEMs and the expression profiles of DEGs were integrated by constructing negative interactions between DEMs and DEGs following methodologies in Cao et al. [35].

## 3. Results

### 3.1. Comparison of Growth Characteristics

Survival rates of tilapia in each stock exceeded 95% after 90 days. Growth trait parameters analyzed by LSD test among stocks (Figure 2) showed that after 45 days, WGR, BLGR, BDGR and SGR were greatest in the HY stock, followed by the NL stock, then the AR stock (*p* < 0.05). The SGR HY value was 1.46× and 1.27× that of AR and NL, respectively. After 90 daysays, values for these four growth characteristics for each stock were similarly ordered, with SGR HY being 1.33× and 1.17× that of AR and NL, respectively. Throughout the 90-day period, HY showed significantly superior values over both parents in growth characteristics (*p* < 0.05).

### 3.2. Hematological Parameter Variation

Hematological parameters for each stock after 45 days and 90 days are presented in Figure 3. For each stock, glucose concentrations were greatest in the HY stock, followed by the NL stock, then the AR stock, with those for AR significantly lower than for HY and NL (*p* < 0.05). Glucose concentrations in the NL and HY stocks did not change significantly between 45 days and 90 days. Cholesterol concentrations in the NL and HY stocks were significantly higher (*p* < 0.05) than in the AR stock after 45 days and 90 days, while there was no significant difference (*p* > 0.05) between the NL and HY stocks. Cholesterol concentrations were slightly higher at day 90 compared to day 45. Concentrations of triglyceride and TP and activities of ALT and AST did not differ significantly among the three stocks after 45 days and 90 days.

### 3.3. Summary of Transcriptomes and miRNAomes

After sequencing quality control, an average of 49.7 million raw reads and 49.3 million clean reads were obtained for each transcriptome sample. The average Q20 of samples was 98.6% (Appendix A). An average of 44.8 million clean reads was mapped to reference genomes (Appendix A). After assembling, 14,252, 12,855, and 13,961 unigenes were identified in NL, AR, and HY stocks, respectively (Appendix A).

An average of 11.6 million raw reads and 11.0 million clean reads were obtained for each miRNAome sample. The average Q20 of samples was 96.8% (Appendix A). Of reads, 67.8% ranged from 21–23 bp in length (Appendix A). After alignment, there were 242, 246, and 260 known miRNAs, with 89, 77, and 99 novel miRNAs identified in the NL, AR, and HY stocks, respectively (Appendix A).

### 3.4. Statistics of Differentially Expressed Genes (DEGs) and miRNAs (DEMs)

Differential expression analysis identified 2561 DEGs and 88 DEMs between NL and AR, 1050 DEGs and 48 DEMs between HY and AR, and 822 DEGs and 27 DEMs between HY and NL, respectively (Figure 4). Compared with the numbers of DEGs and DEMs between HY and parental AR and NL stocks, higher DEG and DEM values occurred between AR and NL. Some 65.4% of DEGs between HY and AR overlapped with those between NL and AR, and some 85.2% of DEMs between HY and AR overlapped with those between NL and AR. There were 67.8% DEGs between HY and NL that overlapped with those between NL and AR, and 79.2% DEMs that overlapped with those between NL and AR. These results indicate that most DEGs and DEMs between HY and parental AR/NL were generated from DEGs and DEMs between NL and AR.

An expression matrix was established based on TPM values of all mRNAs and miRNAs in nine samples from the NL, AR and HY stocks. Hierarchical cluster analysis revealed these nine samples are divided into three subclasses consistent with stocks according to expression matrix results (Figure 5). A distant gene expression pattern was found between the two parental NL and AR stocks, while the gene expression pattern of HY was more like that of NL than AR. The expression of DEMs was almost the same between any two stocks.

According to Rapp et al. [32], the DEGs of HY individuals relative to parents could be classified into 12 possible expression categories. The additive pattern (I, II) generated 1499 DEGs, accounting for 48.4% of all 3097 DEGs (Figure 6). The ELD-NL (III, IV) and ELD-AR (V, VI) patterns generated 855 and 627 DEGs, and accounted for 27.6% and 20.2% of all DEGs, respectively. The ELOD-Up (VII–IX) and ELOD-Down (X–XII) patterns generated 39 and 77 DEGs, respectively; the ELOD pattern accounted for 3.7% of all DEGs, far less than for the ELD pattern. The number of DEGs in pattern III was 1.99× that of pattern IV, accounting for 66.6% of all DEGs in the ELD-NL pattern. Similarly, the number of DEGs in pattern V was 1.18× that of pattern VI, accounting for 54.1% of all DEGs in the ELD-AR pattern. The number of DEGs in the ELOD-Up pattern (X–XII) was 1.97× that of ELOD-Down pattern (VII–IX), accounting for 66.4% of all DEGs in ELOD categories.

DEMs could be classified into seven categories (Figure 7). The additive pattern (I, II) generated 40 DEMs, accounting for 39.2% of all DEMs. The ELD-NL (III, IV) and ELD-AR (V, VI) patterns generated 41 and 20 DEMs, and accounted for 40.2% and 19.6% of all DEMs, respectively. Only one DEM occurred in the ELOD pattern (VII), accounting for 0.98% of all DEMs.

### 3.5. Functional Analysis of NEGs

A total of 1598 NEGs and 62 NEMs were identified (Appendix A). KEGG analysis of NEGs revealed 46 pathways to be significantly enriched (*p* < 0.05) (Figure 8), with most related to “organismal systems (aging, endocrine, digestive, and excretory systems)”, “diseases (cancer, drug resistance, neurodegenerative disease)”, and “metabolism (carbohydrate, lipid, amino acid, vitamins, xenobiotics metabolism)”. NEG enrichment results indicate that hybrids’ gene expression differed from parental average gene expression in aspects of growth, digestion, as well as metabolic, immune, and antioxidant capacities.

There were 33, 11, and 4 significantly enriched pathways (*p* < 0.05) identified from genes in the ELD-NL, ELD-AR, and ELOD categories, respectively (Figure 9). ELD-NL category genes were associated with pathways mainly related to “organismal systems (endocrine and digestive system, regeneration, aging)”, “metabolism (carbohydrate, lipid, amino acid, vitamins, xenobiotics metabolism)”, and “diseases (drug resistance, cancer)” (Figure 9A), including, for example, 11 genes involved in the “insulin signal pathway”, 13 genes involved in “growth hormone synthesis, secretion and action”, 10 genes involved in “fatty acid elongation”, and 20 genes involved in “viral carcinogenesis”(Figure 10A). ELD-AR category genes enriched in pathways mainly related to “organismal systems (digestive system)”, “metabolism (lipid, vitamins, xenobiotics metabolism)”, and “diseases (cancer, drug resistance, neurodegenerative diseases)” (Figure 9B), including, for example, 13 genes involved in “bile secretion”, 7 genes involved in “metabolism of xenobiotics by cytochrome P450”, 10 genes involved in “steroid hormone biosynthesis”, and 10 genes involved in “platinum drug resistance” (Figure 10B). ELOD category genes enriched in pathways mainly related to “lipid metabolism” (Figure 9C), including, for example, 7 genes involved in “steroid hormone biosynthesis” and 6 genes involved in “fatty acid elongation” (Figure 10C).

### 3.6. Interactions between DEMs and DEGs

A total of 264 putative miRNA-mRNA interaction pairs (53 DEMs and 168 DEGs) were predicted. The regulatory networks of miRNA-mRNA interaction pairs were evaluated using Cystoscope v3.9 (Figure 11A). Most DEGs from miRNA-mRNA interactions were up-regulated, accounting for 61.4% of all interaction pairs (Figure 11B). A total of 187 miRNA-mRNA interaction pairs occurred in the category of ELD-NL pairs, accounting for 70.8% of all interaction pairs, followed by the category of ELD-AR pairs, with 77 interaction pairs accounting for 29.2% of all interaction pairs. No miRNA-mRNA interaction pairs were found in the ELOD mode. In negatively regulated miRNA-mRNA pairs, the top 10 enriched pathways annotated in KEGG analysis (Figure 12A) included “organismal systems (endocrine system)”, “metabolism (lipids and amino acids)”, “diseases (cancer and viral infection)”, “cellular processes (cell motility)”, and “environmental information processing (signal transduction)”. Notably, most miRNA-mRNA interaction pairs involved in metabolism of lipids and amino acids were from the category of ELD–NL pairs (Figure 12B), such as oni-miR-204a–*hykk*, oni-miR-10581a–*odc1*, oni-miR-150–*gclm*, and oni-miR-429b-*chst12*. Almost all DEGs from miRNA-mRNA interaction pairs from the category of ELD-NL pairs were up-regulated in hybrids, indicating that these miRNA-mRNA interaction pairs played key roles in the activation of metabolic processes.

## 4. Discussions

### 4.1. Improved Performance in Growth and Hematological Metabolic Indices in Hybrid Tilapia

The fact that many aquatic animals such as fishes and shellfishes produce offspring with improved traits through interspecific hybridization is widely exploited in aquaculture [12,36,37,38]. In crosses of *O. niloticus*
♀ × *O. aureus* ♂, 100% of hybrid tilapia are male [39]. Hybrid offspring combine advantages of the parents, including accelerated growth and greater resistance to low temperatures and streptococcal disease [40,41]. We demonstrated that hybrids of *O. niloticus* and *O. aureus* have significantly improved growth performance over parental stocks, possibly because of interactions between the complementary maternal NL and paternal AR genomes.

Plasma is increasingly collected from fish because it can indicate the nutritional, metabolic and health status of an individual [42,43]. We report TP and triglyceride concentrations and AST and ALT activities to be unaffected by differences in parental stocks. In contrast, the plasma glucose concentrations in HY individuals were higher than those in NL and AR individuals. Glucose concentrations in fish are found positively correlated with growth performance in fishes [44,45], possibly because fish need to mobilize large amounts of energy during growth, with plasma glucose concentrations increasing in faster-growing fish to synthesize proteins in muscles. Cholesterol, an important precursor of physiologically active compounds in fishes, is an important component of membranes, and a precursor of biologically active compounds including bile acids, steroid hormones, and vitamin D [46,47]. We report HY and NL stocks to have significantly higher plasma cholesterol levels than AR stocks, which may be closely related to up-regulated genes enriched in the cholesterol metabolism pathway in the liver of HY and NL fish. Because cholesterol synthesis is generally improved in faster-growing fish [47], an elevated cholesterol concentration in plasma may indicate increased fat digestion and lipid metabolism.

### 4.2. Roles of NEGs and NEMs in Formation of Super-Parental Vigor in Hybrid Tilapia

In hybrids of different species or stocks, the genomes from both parents interact [48,49]. Compared to mammals, fishes are more likely to hybridize between species, so fish hybrids make good models to explore heterosis and hybrid variation. Recent studies of differences in allelic gene expression between interspecific fish hybrids and parents based on transcriptome analysis have attempted to explain the mechanism of superior hybrid traits [12,50,51,52,53]. Non-additive expression represents a new gene expression pattern in hybrids, which is important for the realization of heterosis [54,55,56]. This form of expression results from cis- and trans-regulation between two parental genomes [1,57], with these interactions resulting in significant deviations between hybrid gene expression values and average parental expression values. Non-additive gene expression is common in hybrids of aquatic animals, such as catfish *Pelteobagrus fulvidraco* × *p. vachelli* [15], groupers *Epinephelus fuscogutatus* × *E. lanceolatus* [53], seabreams *Acanthopagrus schlegelii* × *Pagrus major* [12], and two subfamilies of pearl oysters *Pinctada fucata* [58]. These non-additively expressed genes are closely related to growth rate, hypoxia tolerance, and stress resistance in hybrids. Zhou et al. [59] reported 8 of 12 lipid-metabolism-related genes in the intestine of hybrid tilapia to show non-additive expression, including 5 genes for which the expression levels were significantly higher than in parents, suggesting that they played a key role in super-parental growth and lipid utilization advantages. Zhong et al. [19] traced the molecular outcomes of growth hormone expression and allele-specific expression in hybrid tilapia, and found cis- and trans-regulation promoted the expression of maternal genomes in hybrids, resulting in significantly higher levels of growth hormone in hybrids than in parents.

It was reported that 1598 NEGs out of a total of 3097 DEGs were identified in hybrids, indicating that interspecific hybridization causes significant changes in the process of gene transcription. These NEGs may be closely related to heterosis in growth, metabolism, extreme environmental adaptation, and disease resistance in hybrid tilapia. We also identified 67 NEMs out of 120 DEMs, with NEMs accounting for more than half of all DEMs (similar in proportion to the ratio of NEGs to DEGs), indicating that miRNA in hybrids has a significant regulatory effect in post-transcriptional processes. Recent studies report NEMs to play a key role in the dominant expression of functional genes in hybrids [11,60]. By identifying negative interactions between DEMs and DEGs, we found 32 known NEMs and 21 novel NEMs in miRNA-mRNA interaction pairs. Although studies on miRNAs in fishes are few, several studies have reported miRNA-mRNA pairs to be closely related to growth, metabolism, and immunity functions [61,62,63], suggesting that NEM regulation plays an important role in the formation of super-parental advantages in growth rate and environmental adaptation of hybrid tilapia.

### 4.3. The Significance of Gene ELD and ELOD Patterns

For a given non-additively expressed gene, if the total expression in hybrids is statistically identical to that of a diploid parent, we refer to this as ‘expression level dominance’ (ELD). If the expression is beyond the range of both parents, we refer to this as ‘expression level overdominance’ (ELOD) [64]. Many functional genes show ELD or ELOD patterns in fishes, such as hybrids of brook charr (*Salvelinus fontinalis*) [65], pufferfish (*Takifugu rubripes* × *T. flavidus*) [13], and grouper (*E. fuscogutatus* × *E. lanceolatus*) [53]. Similarly, more than half of the DEGs in our HY stock presented ELD or ELOD patterns.

Rapp et al. [32] first divided non-additive expressed genes in cotton (*Gossypium*) hybrids into 10 subgroups, providing new insights into the architecture of gene expression in the allopolyploid nucleus. We report the number of NEGs with ELD patterns (III–VI) to be much higher than that with ELOD patterns (VII–XII). More than 60% of NEGs from ELD patterns were up-regulated in HY individuals, indicating that HY fish integrated the transcriptional predominance from parental DEGs. Similar results were found in hybrid yellow catfish and hybrid cyprinids [15,37]. We posit that novel gene expression patterns in the HY stock might be the result of dominant allele expression complementation from parents. Similar expression models have been reported for plant hybrids, such as those of oilseed rape [11] and cotton [30]. ELOD-Up pattern genes represented 66.4% of all DEGs in ELOD categories. This indicates that overdominance is a unique gene regulatory interaction that achieves heterosis in hybrids [66,67].

In KEGG analysis, most ELD and ELOD genes and miRNA-mRNA pairs were enriched in pathways related to organismal systems, metabolism, cellular processes, and diseases. Considering phenotypic heterosis of cultured HY, we divided these differentially expressed genes into three functional categories: growth and development, metabolism, and immunity and antioxidant capacity (Section 4.3.1, Section 4.3.2, Section 4.3.3).

#### 4.3.1. Growth and Development

In aquatic animals, ELD or ELOD genes are often related to growth and development [6,68,69]. We report HY individuals to be NL-dominant in endocrine- and digestive system-related pathways such as “insulin signal pathway”, “growth hormone synthesis, secretion and action”, “carbohydrate digestion and absorption”, and “fat digestion and absorption”, and AR-dominant and overdominant in the digestive system-related pathway “bile secretion”. These results indicate that dominant or overdominant genes are involved in the endocrine and digestive systems and work together to yield heterosis of growth and development in HY tilapia. In addition, it is notable that many NL- or AR-dominant genes were enriched in cancer-related pathways such as “viral carcinogenesis”, “chemical carcinogenesis”, “thyroid cancer”, and “bladder cancer”. Although it is not clear if these cancer-related genes are involved in growth and development functions in fish, similar cancer-related pathway genes were found to be associated with cell proliferation and development [70]. Similar cancer genes are involved in heterosis in hybrid yellow catfish [15], and these dominant cancer genes may play important roles in heterosis of growth and development in hybrid tilapia.

#### 4.3.2. Metabolism

In aquatic animals, differences in the expression of genes related to nutrient and energy metabolism also play key roles in the formation of super-parental vigor in hybrid growth performance and stress tolerance [14,29,59,71]. Accordingly, the majority of studies have examined differences in metabolic activities between hybrids and parents. We report lipid metabolism pathways to be mostly enriched by ELD-NL and ELOD genes, including “fatty acid elongation”, “biosynthesis of unsaturated fatty acids”, and “primary bile acid biosynthesis”. Vitamin and amino acid metabolic pathways are mostly enriched by ELD-NL and ELD-AR genes, including “retinol metabolism” and “porphyrin and chlorophyll metabolism”. Carbohydrate metabolism pathways are mostly enriched by ELD-NL genes, including “pentose and glucuronate interconversions” and “ascorbate and aldarate metabolism”. All of these pathways are crucial for fish to improve their growth rates and stress tolerance [13]. ELD-NL and ELD-AR miRNA-mRNA pairs were also enriched in pathways related to metabolism of lipids and amino acids. These results overlapped with results of analysis of ELD gene pathways, indicating that gene expression variations in HY fish nutritive metabolic processes are regulated by non-additive miRNAs. These interacting miRNA-mRNA pairs may promote nutrient assimilation processes in hybrids.

#### 4.3.3. Immunity and Antioxidant Capacity

In aquatic animals, hybrids are often cited as having increased immunity and antioxidant capacity compared to parents, and as having greater hybrid vigor with regard to infectious bacterial and viral resistance and adaptation to hypoxia and ammonia nitrogen [6,12,72]. This heterosis could improve the survival rate of hybrids, which is important for aquaculture. However, compared to shrimp, mollusks, and other aquatic animals [73,74], the molecular mechanisms of heterosis in immunity and antioxidant capacity in fish hybrids remain unclear. We report disease and cellular process pathways to be mostly enriched from ELD-NL and ELD-AR genes, including “apoptosis”, “hepatitis C”, “p53 signaling pathway”, and “platinum drug resistance”. ELD-NL and ELD-AR miRNA-mRNA pairs were also found to be enriched in the “human papillomavirus infection” pathway. Although these ELD gene pathways and ELD miRNA-mRNA pair pathways are classified in the diseases and cellular processes subclass, they play a key role in activating non-specific immunity in fishes [75,76]. ELD-NL and ELD-AR genes were also enriched in the “drug metabolism-cytochrome P450” and “metabolism of xenobiotics by cytochrome P450” pathways. Although these pathways are in the xenobiotic biodegradation and metabolism subclass, they are involved in improving antioxidant capacity in hybrid fishes [15,37]. We conclude that HY fish inherited their excellent tolerance of oxidative stress from maternal NL and paternal AR through related dominant allele expression complementation. These results provide evidence that the dominant expression of parental genes related to immune defense and antioxidant capacity is integrated in the heterosis of hybrid individuals.

## 5. Conclusions

The growth performance and hematologic biochemical parameters of hybrid and parental tilapias were compared, and growth characteristics and hematological glucose and cholesterol parameters were reported to be significantly improved in hybrids. The mRNA and miRNA transcriptional analyses of two stocks of tilapia and their hybrids were investigated by using high-throughput sequencing (Figure 13). Differential expression analysis identified 3097 DEGs and 120 differentially expressed miRNAs (DEMs) among these three stocks, in which 1598 DEGs and 62 DEMs were non-additively expressed (NEGs and NEMs) in hybrids. Both ELD and ELOD patterns occurred in DEGs and DEMs, indicating that dominance and overdominance models were widespread in the transcriptional and post-transcriptional regulation of genes involved in growth, metabolism, immunity, and antioxidant capacity in hybrid tilapia. Our potential negative regulatory networks between DEMs and predicted target DEGs determined that most DEGs from miRNA–mRNA interactions were up-regulated. Dominance and overdominance models in the transcriptome and miRNAome facilitated integration of advantageous parental alleles into hybrids, contributing to their improved growth and survival. The present study provides new insights into molecular heterosis in tilapia, which may contribute to revealing the mechanisms underlying this complex phenomenon in other aquatic animals.

## Figures and Tables

**Figure 1 animals-12-00640-f001:**
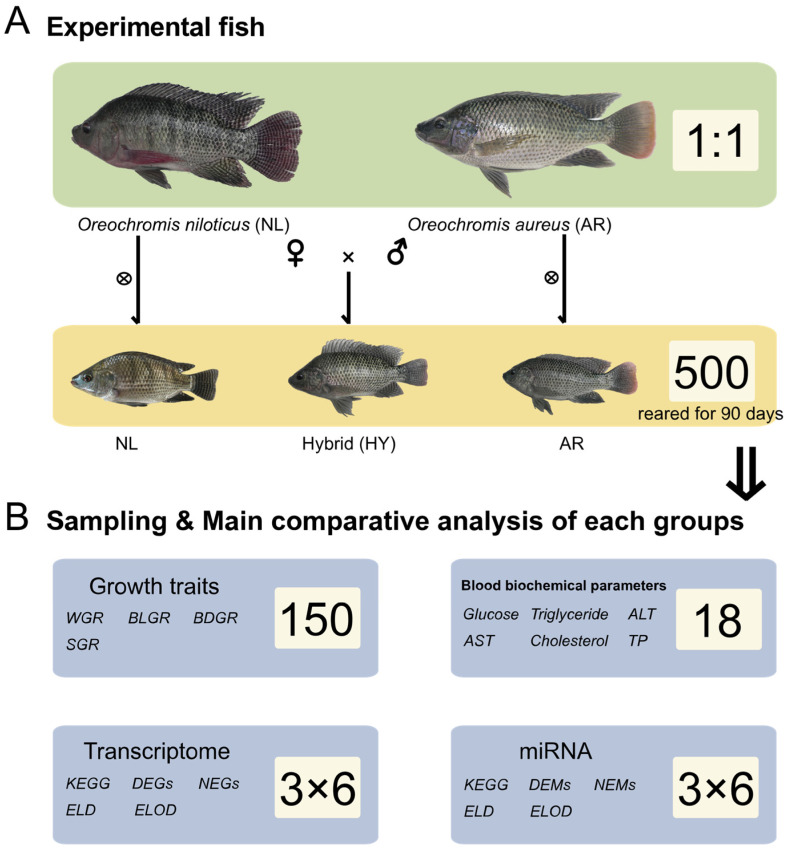
Experimental design and workflow.

**Figure 2 animals-12-00640-f002:**
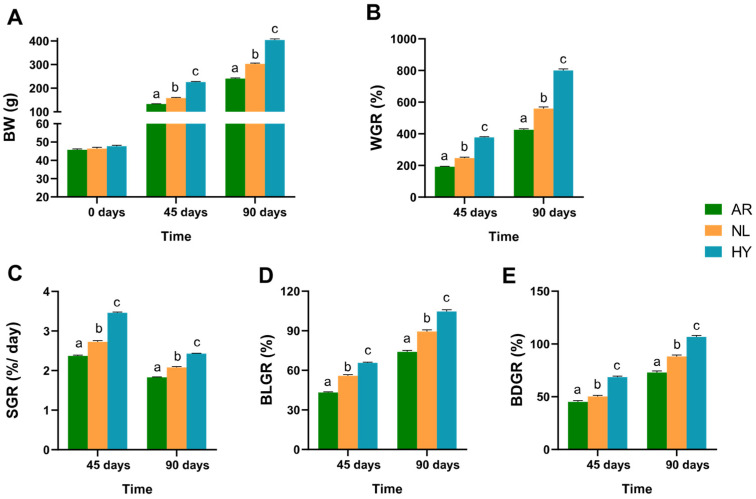
Variation in growth characteristics in *O. niloticus* (NL), *O. aureus* (AR) and hybrid (HY) stocks after 45 days and 90 days. (**A**) Variation in body weight (BW) in NL, AR and HY stocks. (**B**) Variation in WGR in NL, AR and HY stocks. (**C**) Variation in SGR in NL, AR and HY stocks. (**D**) Variation in BLGR in NL, AR and HY stocks. (**E**) Variation in BDGR in NL, AR and HY stocks. Values are presented by a, b and c as mean ± SD.

**Figure 3 animals-12-00640-f003:**
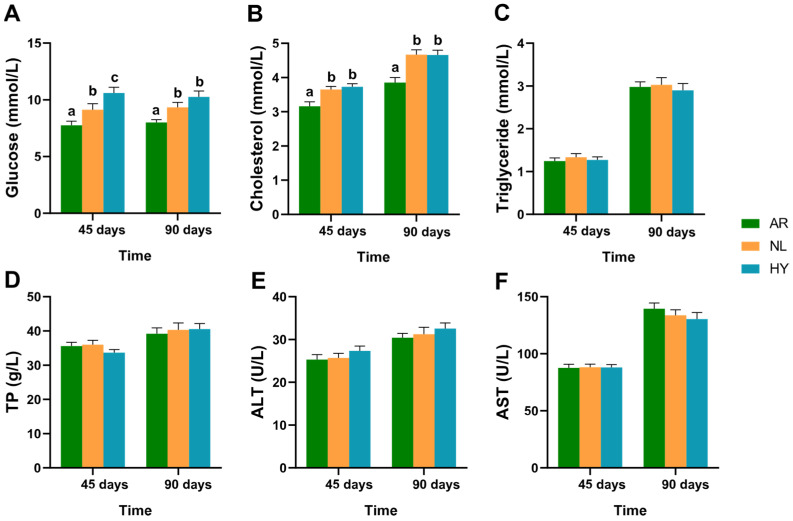
Hematologic biochemical parameters of *O. niloticus* (NL), *O. aureus* (AR) and hybrid (HY) stocks after 45 days and 90 days. (**A**) Variation in glucose concentrations in NL, AR and HY stocks. (**B**) Variation in cholesterol concentrations in NL, AR and HY stocks. (**C**) Variation in triglyceride concentrations in NL, AR and HY stocks. (**D**) Variation in TP concentrations in NL, AR and HY stocks. (**E**) Variation in ALT activities in NL, AR and HY stocks. (**F**) Variation in AST activities in NL, AR and HY stocks. Values are presented by a, b and c as mean ± SD.

**Figure 4 animals-12-00640-f004:**
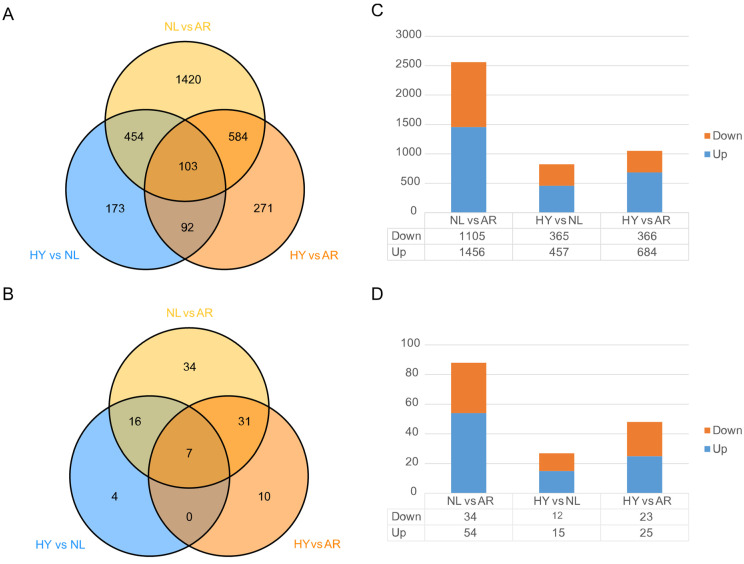
Differential expression of genes and miRNAs in *O. niloticus* (NL), *O. aureus* (AR) and hybrid (HY) stocks. Venn diagrams of DEGs (**A**) and DEMs (**B**), and histograms of DEGs (**C**) and DEMs (**D**).

**Figure 5 animals-12-00640-f005:**
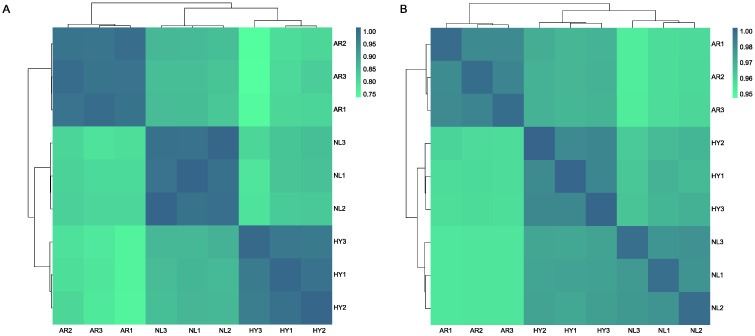
Heatmap clustering of the (**A**) expressed genes and (**B**) miRNAs in *O. niloticus* (NL), *O. aureus* (AR) and hybrid (HY) stocks.

**Figure 6 animals-12-00640-f006:**
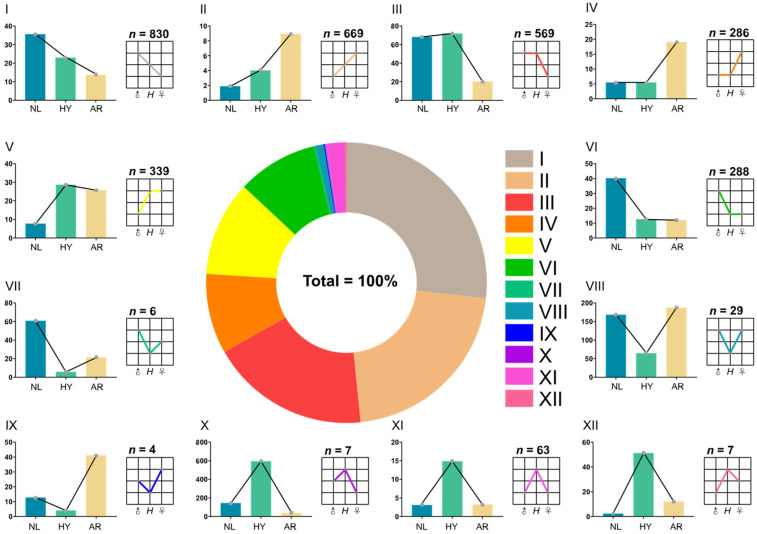
Patterns in differential expression of DEGs in *O. niloticus* (NL), *O. aureus* (AR) and hybrid (HY) stocks. Roman numerals **I**–**XII** represent different gene expression patterns.

**Figure 7 animals-12-00640-f007:**
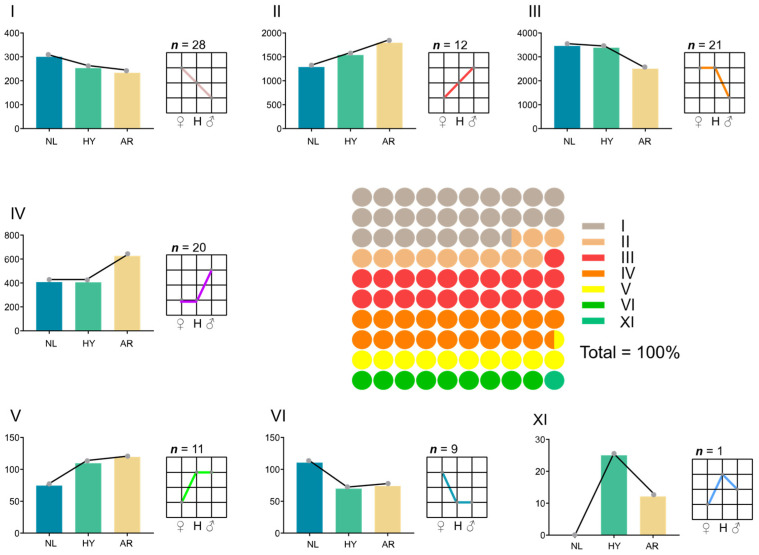
Differentially expressed patterns in DEMs of *O. niloticus* (NL), *O. aureus* (AR) and hybrid (HY) stocks. Roman numerals (**I**–**VI**), **XI** represent different miRNA expression patterns.

**Figure 8 animals-12-00640-f008:**
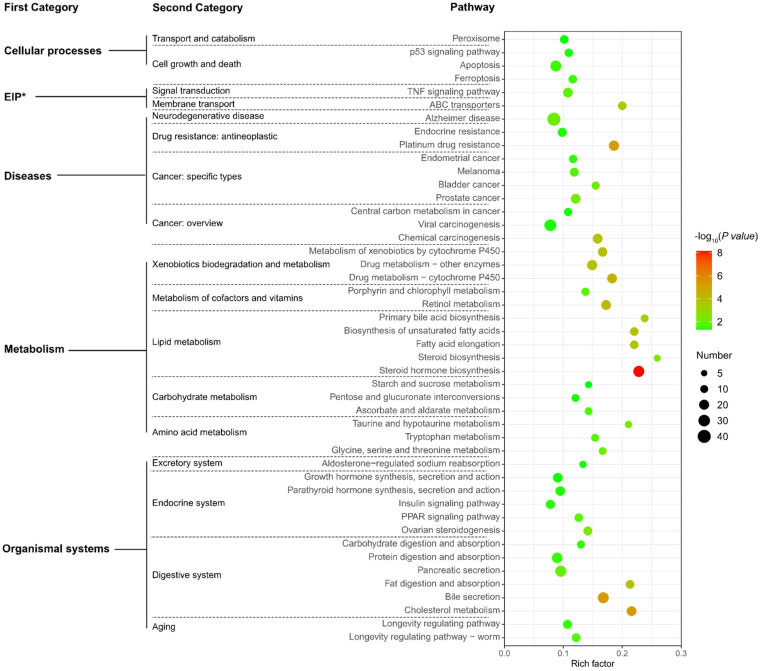
Analysis of KEGG pathway enrichment in NEGs. Note: EIP*, environmental information processing.

**Figure 9 animals-12-00640-f009:**
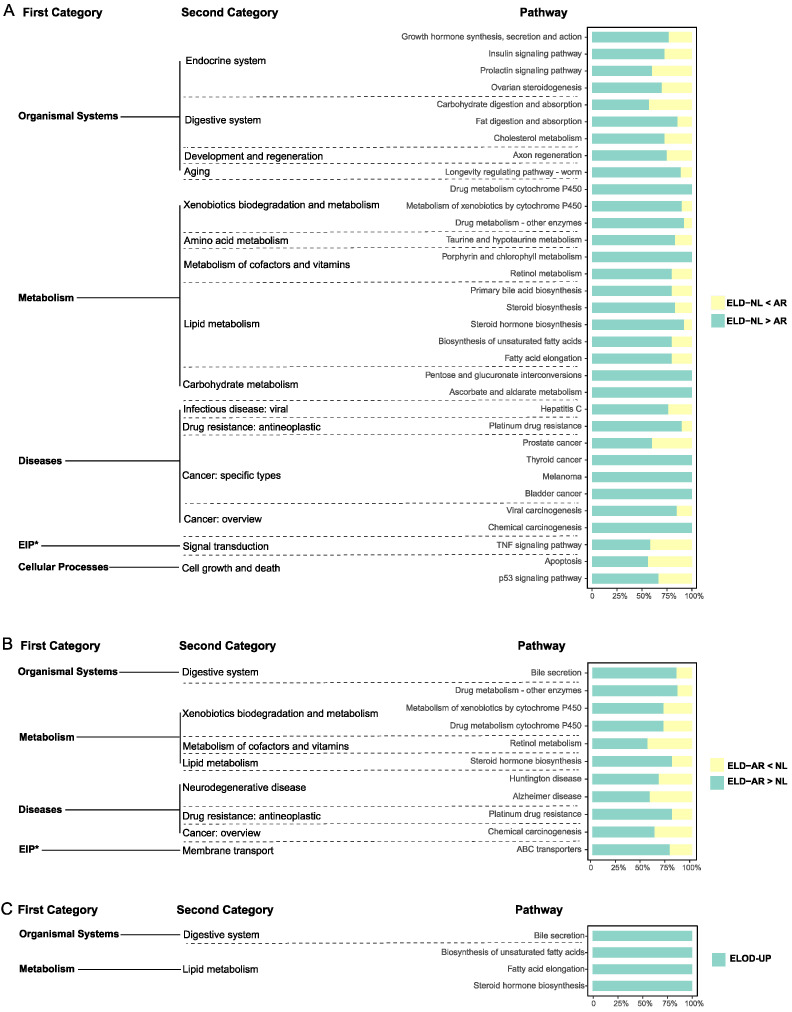
Analysis of KEGG pathway enrichment for ELD-NL (**A**), ELD-AR (**B**) and ELOD genes (**C**). Note: EIP* denotes environmental information processing.

**Figure 10 animals-12-00640-f010:**
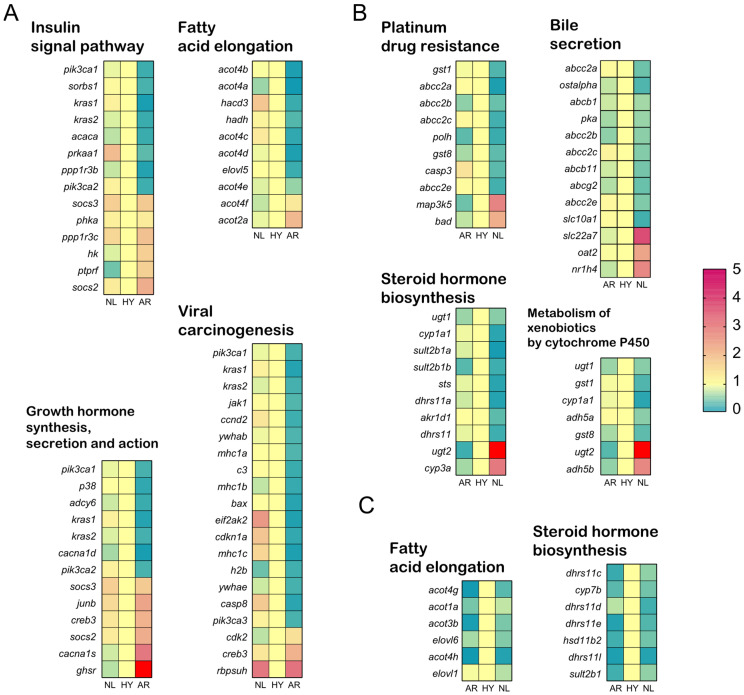
Relative expression of genes in various enrichment pathways from ELD-NL (**A**), ELD-AR (**B**) and ELOD (**C**) in *O. niloticus* (NL), *O. aureus* (AR) and hybrid (HY) stocks.

**Figure 11 animals-12-00640-f011:**
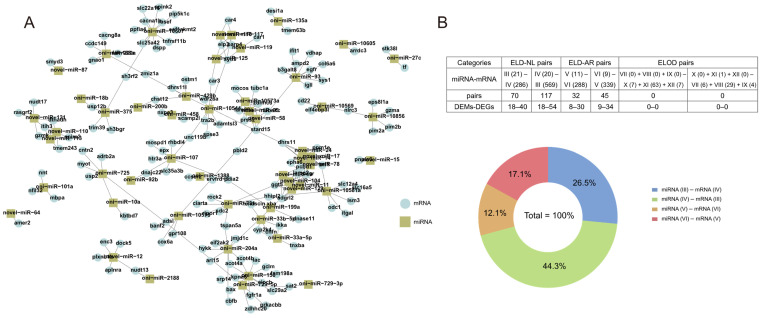
Overview of predicted negative miRNA-mRNA interaction pairs. (**A**) Integrated biomolecular negative interaction network of miRNA-mRNA based on 12 expression patterns. (**B**) Statistics for predicted negative miRNA-mRNA interaction pairs based on 12 expression patterns.

**Figure 12 animals-12-00640-f012:**
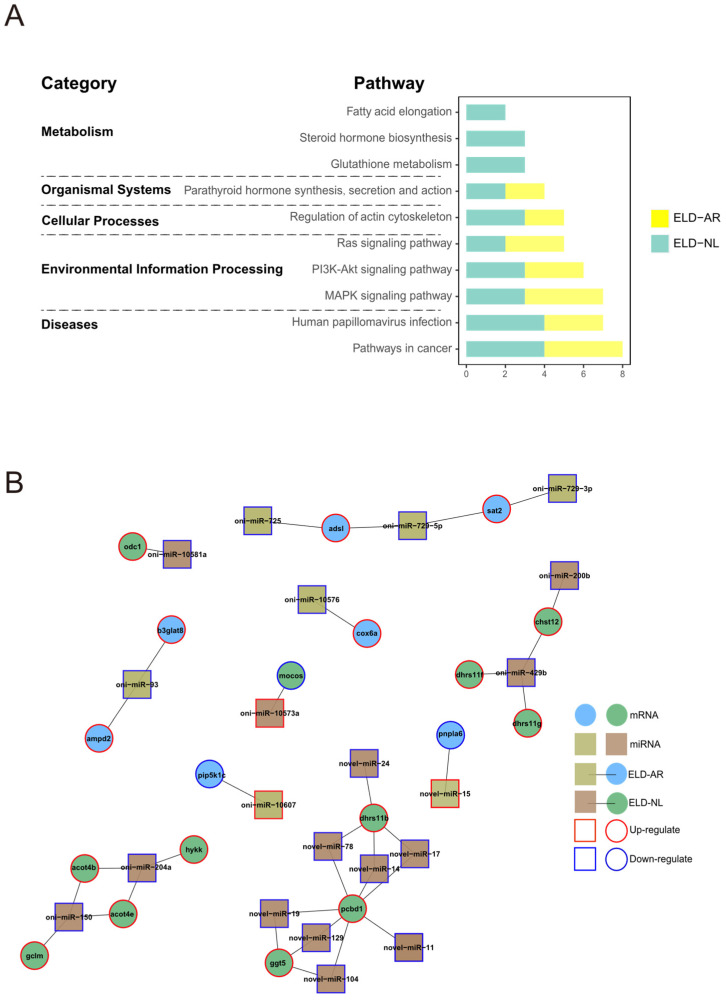
Analysis of KEGG pathway enrichment of predicted negative miRNA-mRNA interaction pairs. (**A**) KEGG pathway enrichment of predicted negative miRNA-mRNA interaction pairs. (**B**) Integrated negative interaction network of miRNA-mRNA pairs involved in metabolism.

**Figure 13 animals-12-00640-f013:**
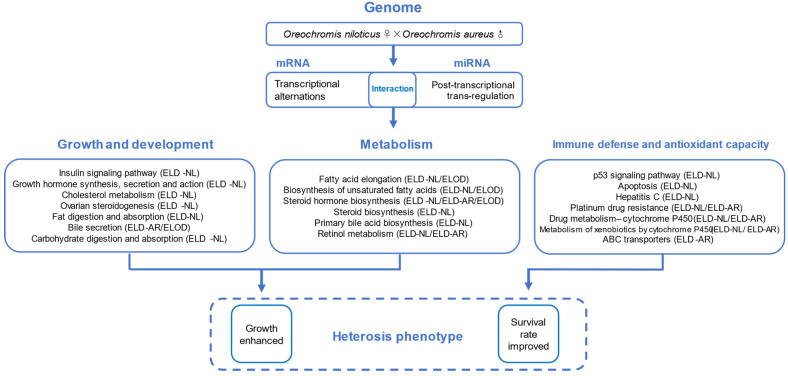
Hypothetical model of heterosis in *O. niloticus* × *O. aureus* hybrid tilapia.

## Data Availability

The RNA-Seq data have been submitted to the Sequence Read Archive (SRA) database (No. PRJNA 787752).

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
