# Peer review of "Integration of mRNA and miRNA Profiling Reveals Heterosis in Oreochromis niloticus × O. aureus Hybrid Tilapia"

_animals, 2022, doi:10.3390/ani12050640_

Round 1
Reviewer 1 Report
The manuscript of the title “Integration of mRNA and miRNA profiling reveals heterosis in Oreochromis niloticus × O. aureus hybrid tilapia” provides real breeding in a lab to show the hybrid vigor of tilapia and supports the results of hybrid culture with biochemical and in silico molecular studies. Although the in silico investigations have yet to be confirmed by in vitro studies, it is sufficient to publish this early information. There are minor suggestions to improve the manuscript as following.
The author may change a bit of paragraph 2 in the introduction, and it could be linked with paragraph 3.
Figure 4C and 4D: In the table header, please change F to NL, M to AR, and H to HY, or the author may change them to NL (F), AR (M), and HY to correspond with other figures that use NL, AR, and HY.
Figure 11A and 12B: There is no indication of circular nodes and square nodes, whether they represent mRNAs or miRNAs in legend figure. Please describe more details of colors, shapes, and lines to represent the figure.
Reviewer 2 Report
While the heterosis of certain hybrids, including hybrid fishes, has long been recognized, the mechanisms giving rise to heterosis are not well understood. Early explanations involved superior function of hybrid molecules derived from the respective parents, but recent one focus on differential gene expression in hybrids relative to the parental species. One heterotic hybrid results from the crossing of Nile and blue tilapias, and Xiao et al. compare of growth, hematological parameters, gene expression and small RNAs among the parental species and the hybrid. While I have some issues here and there, this study is a technical coup, and has interest beyond the tilapias at issue. I relate a few context-related issues here. To help polish the prose, I have marked the manuscript. Of note, more than one species of fish is expressed as fishES.
Summary and Abstract. – At line 16, miRNA should be defined parenthetically for the readership of this journal.
Introduction. – At line 69, not all crosses of these two species yield 100% males (a review by Gideon Hulata makes that clear), so an appropriate qualifier should be added.
Methods. – I’ve penned suggested minor revisions on the manuscript document.
Results. – I’ve penned suggested minor revisions on the manuscript document.
Discussion. – Again, I’ve penned suggested minor revisions on the manuscript document. Parts of the Conclusion referring to results obtained in this study should be presented in the past tense.
References. – I’ve marked departures from journal citation stylistics.

Reviewer 3 Report
In my opinion, a high-quality manuscript. Perfectly planned and conducted research. My few remarks are contained in the text of MS. To see them all, open the file in Acrobat Reader. I recommend the authors to supplement the legends of figures.
